# Brain-Derived Major Glycoproteins Are Possible Biomarkers for Altered Metabolism of Cerebrospinal Fluid in Neurological Diseases

**DOI:** 10.3390/ijms24076084

**Published:** 2023-03-23

**Authors:** Kyoka Hoshi, Mayumi Kanno, Aya Goto, Yoshikazu Ugawa, Katsutoshi Furukawa, Hiroyuki Arai, Masakazu Miyajima, Koichi Takahashi, Kotaro Hattori, Keiichi Kan, Takashi Saito, Yoshiki Yamaguchi, Takashi Mitsufuji, Nobuo Araki, Yasuhiro Hashimoto

**Affiliations:** 1Department of Biochemistry, Fukushima Medical University, Fukushima City 960-1295, Fukushima, Japan; 2Department of Forensic Medicine, Fukushima Medical University, Fukushima City 960-1295, Fukushima, Japan; 3Center for Integrated Science and Humanities, Fukushima Medical University, Fukushima City 960-1295, Fukushima, Japan; 4Department of Human Neurophysiology, Fukushima Medical University, Fukushima City 960-1295, Fukushima, Japan; 5Division of Community Medicine, Tohoku Medical and Pharmaceutical University, Sendai 981-8558, Miyagi, Japan; 6Institute of Development, Aging and Cancer, Tohoku University, Sendai 980-8575, Miyagi, Japan; 7Department of Neurosurgery, Juntendo University, Bunkyo City 113-8421, Tokyo, Japan; 8Department of Neurosurgery, Sanno Hospital, Minato City 107-0052, Tokyo, Japan; 9Department of Mental Disorder Research, National Institute of Neuroscience, National Center of Neurology and Psychiatry (NCNP), Kodaira 187-8502, Tokyo, Japan; 10Department of Anesthesiology, Southern Tohoku General Hospital, Koriyama 963-8052, Fukushima, Japan; 11Department of Neurocognitive Science, Nagoya City University, Nagoya 467-8601, Aichi, Japan; 12Laboratory of Pharmaceutical Physical Chemistry, Tohoku Medical and Pharmaceutical University, Sendai 981-8558, Miyagi, Japan; 13Department of Neurology, Saitama Medical University Hospital, Koshigaya 350-0495, Saitama, Japan

**Keywords:** cerebrospinal fluid, choroid plexus, neuron, prostaglandin D2 synthase, brain-derived transferrin, neurological diseases

## Abstract

Cerebrospinal fluid (CSF) plays an important role in the homeostasis of the brain. We previously reported that CSF major glycoproteins are biosynthesized in the brain, i.e., lipocalin-type prostaglandin D2 synthase (L-PGDS) and transferrin isoforms carrying unique glycans. Although these glycoproteins are secreted from distinct cell types, their CSF levels have been found to be highly correlated with each other in cases of neurodegenerative disorders. The aim of this study was to examine these marker levels and their correlations in other neurological diseases, such as depression and schizophrenia, and disorders featuring abnormal CSF metabolism, including spontaneous intracranial hypotension (SIH) and idiopathic normal pressure hydrocephalus (iNPH). Brain-derived marker levels were found to be highly correlated with each other in the CSF of depression and schizophrenia patients. SIH is caused by CSF leakage, which is suspected to induce hypovolemia and a compensatory increase in CSF production. In SIH, the brain-derived markers were 2–3-fold higher than in other diseases, and, regardless of their diverse levels, they were found to be correlated with each other. Another abnormality of the CSF metabolism, iNPH, is possibly caused by the reduced absorption of CSF, which secondarily induces CSF accumulation in the ventricle; the excess CSF compresses the brain’s parenchyma to induce dementia. One potential treatment is a “shunt operation” to bypass excess CSF from the ventricles to the peritoneal cavity, leading to the attenuation of dementia. After the shunt operation, marker levels began to increase within a week and then further increased by 2–2.5-fold at three, six, and twelve months post-operation, at which point symptoms had gradually attenuated. Notably, the marker levels were found to be correlated with each other in the post-operative period. In conclusion, the brain-derived major glycoprotein markers were highly correlated in the CSF of patients with different neurological diseases, and their correlations were maintained even after surgical intervention. These results suggest that brain-derived proteins could be biomarkers of CSF production.

## 1. Introduction

Cerebrospinal fluid (CSF) plays a pivotal role in the homeostasis of the central nervous system by distributing growth factors, removing metabolic waste, controlling pH, and maintaining intracranial pressure (ICP) [1]. Cushing hypothesized that the CSF is produced mainly by the choroid plexus of the lateral ventricles of the brain, and is absorbed in the arachnoid granulations (bulk flow theory) [2]. In this way, CSF in the lateral ventricles passes through the interventricular foramina to the third ventricle, then from the cerebral aqueduct to the fourth ventricle. The fluid then passes from the fourth ventricle into the subarachnoid space through the foramina of Magendie and Luschka. CSF in the subarachnoid space bathes the surface of the brain and is subsequently absorbed by arachnoid granulations near the parietal bone. The bulk flow theory is accepted particularly by neurosurgeons as it adequately explains the mechanism underlying obstructive hydrocephalus; i.e., aqueduct stenosis blocks CSF flow, causing upstream compartments to become enlarged due to CSF accumulation. In contrast, Jessen et al., hypothesized that the “glymphatic system” plays a major role in generating the CSF flow; i.e., body fluid in para-arterial spaces enters the brain parenchyma, it collects waste, and returns to lymphatic systems via para-venous spaces [3,4]. The mechanism of CSF production and absorption thus remains contentious.

The composition of major proteins in the CSF is similar to that of blood [5,6]. For instance, CSF and serum contain albumin as a major component, constituting 50–60% of their total protein. Albumin is biosynthesized only in the liver, indicating that it is secreted from the liver into blood and then translocated to the CSF. Notably, the protein concentration in plasma/serum is approximately 60–80 mg/mL, which is 50–100 times higher than in the CSF [1,7], suggesting that a small amount of serum protein diffuses into the CSF possibly down its concentration gradient. In this way, the “blood–brain barrier” is leaky even under physiological conditions.

Iron plays many important roles in the brain, and has involvement in myelination, neurotransmission and electron transfer in the respiratory chain [8]. However, as free iron results in cellular toxicity by generating harmful reactive oxygen species (ROS) [9], a physiological response has evolved whereby iron binds to specific carrier proteins, such as transferrin (Tf), in the extracellular fluid and blood [10]. Like albumin, Tf that is glycosylated with sialylated glycans (Sia-Tf) is biosynthesized in the liver, secreted into the blood, and then translocated to the CSF. We also found that isoforms of Tf that carry unique glycans were produced in the brain [11]: these include Tf that carry N-acetylglucosamine-terminated glycans (GlcNAc-Tf) and Tf that carry mannose-terminated glycans (Man-Tf). These Tf isoforms were undetectable in the CSF of hydranencephaly patients, who congenitally lack most of the cerebrum [12], indicating that they are produced by the cerebrum. To identify Tf-producing cells in the cerebrum, we analyzed the expression of Tf mRNA by in situ hybridization [11]. The most intense signals were detected in the choroid plexus epithelium stained with anti-Tf antibody, as well as a GlcNAc-binding probe, recombinant *Psathyrella velutina* lectin (rPVL) [13]. Co-localized signals were found in the epithelium, suggesting that GlcNAc-Tf is produced in the choroid plexus epithelium. In situ hybridization also revealed that Tf mRNA was highly expressed in most neurons in the cerebral cortex, hippocampus, and basal ganglia. To identify glycan isoforms expressed in the cerebral cortex, Tf was purified from the cortex and its glycan content was analyzed by mass spectrometry [11]. Cortical Tf mainly carried mannose-terminated glycans (~90%), suggesting that Man-Tf is secreted by neurons. We speculated, therefore, that Man-Tf could be a biomarker for neurons and that its expression might be altered in neurodegenerative disorders. Man-Tf levels were quantified in the CSF of patients with Alzheimer’s disease (AD), mild cognitive impairment (MCI), other tauopathies, or synucleinopathies. Man-Tf was significantly increased in AD and MCI but not in the other diseases, indicating that Man-Tf could be a diagnostic marker for AD and MCI [11].

Another major glycoprotein in the CSF is lipocalin-type prostaglandin D2 synthase (L-PGDS). Urade et al., reported that L-PGDS is biosynthesized by the leptomeninges and the choroid plexus, which are CSF-producing tissues [14]. The molecule is a member of a superfamily of lipocalins that have diverse physiological functions, including binding to and transporting lipophilic compounds, inducing sleep, etc. [15]. One unexpected finding was that L-PGDS levels in the CSF correlated well with those of Man-Tf and GlcNAc-Tf in AD, although their origins were different [6]. In addition, the markers were also correlated with each other in other neurodegenerative disorders, tauopathies and synucleinopathies. In the present study, we examined the correlation of these markers in neurological diseases such as depression and schizophrenia, and disorders associated with altered CSF metabolism, i.e., idiopathic normal pressure hydrocephalus (iNPH) and spontaneous intracranial hypotension (SIH).

SIH is caused by the leakage of CSF, which induces CSF loss and intracranial hypotension [16,17,18]. The hypotension often results in brain dislocation, particularly in the standing posture, resulting in a severe headache. CSF leakage is quantified by radiological assessment [19,20], where the ^111^indium-labelled radioisotope (RI) diethylenetriamine penta-acetic acid (^111^In-DTPA) is injected into the CSF and its decline is followed over time by RI cisternography. A considerable amount of radioactivity drains from the CSF into the blood and is then excreted via the urine. Some of the radioactivity is retained in the CSF, detected in the form of residual RI radioactivity. More than 30% of the radioactivity is maintained in the CSF of the control patients 24 h after RI injection; this is compared to SIH patients, where less than 10% remains, thus attesting to the CSF leakage [21]. This leakage is thought to induce hypovolemia and a compensatory production of CSF.

Another abnormality showing altered CSF metabolism is iNPH, which is a senile hydrocephalus that induces dementia and gait disturbance, among other symptoms. [22,23]. The accumulation of CSF in the ventricles is observed as “ventriculomegaly” on MRI imaging, with the excess CSF compressing the brain parenchyma, leading to dementia. This accumulation of CSF is thought to cause a reduction in CSF production. The treatment required to cure iNPH is a “shunt operation”, in which CSF is bypassed from the ventricles to the peritoneal cavity [24], providing an attenuation of symptoms.

In the present study, we quantified the levels of brain-derived glycoproteins in various neurological diseases and found that their levels were well correlated with each other across different pathophysiological conditions.

## 2. Results

### 2.1. Marker Levels in CSF from Patients with Neurological Diseases

We previously demonstrated that several CSF glycoproteins, such as Man-Tf, GlcNAc-Tf, and L-PGDS, were derived from the cerebrum and were correlated with each other in neurodegenerative disorders, including AD, other tauopathies, and synucleinopathy [6,11]. In the present study, we examined marker correlations in neurological diseases such as depression, schizophrenia, SIH, non-SIH, iNPH, and non-iNPH.

Prior to the correlation analysis, we quantified levels of Man-Tf, GlcNAc-Tf, and L-PGDS as brain-derived markers (Figure 1). We also quantified the CSF total protein, which was mainly composed of blood-derived proteins (~80%) such as albumin, IgG, and serum Tf [1,6]. Serum Tf carries sialic acid-terminated glycans (Sia-Tf). Tf isoforms were quantified by a lectin/antibody-sandwich ELISA; the glycan-specific lectins used were *Sambucus sieboldiana* agglutinin (SSA) for Sia-Tf, recombinant *Burkholderia cenocepaciar* (rBC2L-A) for Man-Tf, and rPVL for GlcNAc-Tf. Patients were classified into two groups based on the peak ages of onset: one group included depression, schizophrenia, SIH, and non-SIH patients with peak ages in the thirties and forties; another group included iNPH and non-iNPH patients with peak ages in the seventies. The two groups were analyzed statistically, avoiding the potential confounding effects of aging. In depression and schizophrenia patients, the levels of all markers were slightly higher than those of non-SIH patients, but the differences were not significant. SIH patients showed very high and variable levels of all markers compared to non-SIH patients. Man-Tf, GlcNAc-Tf, L-PGDS, Sia-Tf, and total protein levels were increased by 2.9-, 2.2-, 2.0-, 1.7-, and 1.6-fold, respectively, indicating that both brain- and blood-derived proteins were markedly increased in SIH. In iNPH, GlcNAc-Tf and L-PGDS levels were significantly higher than the levels in non-iNPH patients.

### 2.2. CSF Markers in Psychiatric Disease Patients

#### 2.2.1. Correlation of CSF Markers in Patients with Depression

We analyzed correlations between CSF markers in CSF samples from patients with depression. Scatter diagrams and Spearman’s rank correlation coefficients are indicated for each marker combination (Figure 2, Table 1). Brain-derived markers were highly correlated with each other: Man-Tf vs. GlcNAc-Tf (rs = 0.87), Man-Tf vs. L-PGDS (rs = 0.77), and GlcNAc-Tf vs. L-PGDS (rs = 0.70). In contrast, Man-Tf was not correlated with Sia-Tf (rs = 0.20) or total protein (rs = 0.16). Nevertheless, Sia-Tf and total protein were well correlated with each other (rs = 0.65).

#### 2.2.2. Correlations between CSF Markers in Schizophrenia Patients

Correlations between CSF markers were analyzed in schizophrenia patients. Scatter diagrams and Spearman’s rank correlation coefficients are indicated in Figure 3 and Table 2 for each marker combination. The following brain-derived markers showed correlations with each other: Man-Tf vs. GlcNAc-Tf (rs = 0.76), Man-Tf vs. L-PGDS (rs = 0.68), and GlcNAc-Tf vs. L-PGDS (rs = 0.55). In contrast, Man-Tf was not correlated with Sia-Tf (rs = 0.13) or total protein (rs = 0.18).

### 2.3. CSF Markers in SIH

#### 2.3.1. Correlation of CSF Markers in CSF of SIH Patients

SIH is caused by CSF leakage, which induces intracranial hypotension and the dilatation of veins. An increase in CSF production may occur to compensate for this leakage. All marker levels were elevated and varied widely in CSF samples from SIH patients (Figure 4). In spite of the extreme variability of values, brain-derived marker levels were significantly correlated with each other (Table 3): Man-Tf vs. GlcNAc-Tf (rs = 0.57), Man-Tf vs. L-PGDS (rs = 0.45), and GlcNAc-Tf vs. L-PGDS (rs = 0.47). Except for Man-Tf, these markers correlated with the blood-derived marker levels.

#### 2.3.2. Correlation of Markers in CSF from Non-SIH Patients

CSF marker correlations were analyzed in CSF samples from non-SIH patients. Scatter diagrams and Spearman’s rank correlation coefficients are indicated for marker combinations (Figure 5, Table 4). Except for GlcNAc-Tf vs. L-PGDS, the brain-derived markers correlated significantly with each other: Man-Tf vs. GlcNAc-Tf (rs = 0.79), Man-Tf vs. L-PGDS (rs = 0.55). In addition, Sia-Tf and total protein were moderately correlated with the following markers: Man-Tf (rs = 0.51~0.44), GlcNAc-Tf (rs = 0.46~0.50), and L-PGDS (rs = 0.51~0.67). Sia-Tf and total protein were also well correlated (rs = 0.73).

### 2.4. CSF Markers in iNPH

#### 2.4.1. Correlation of Markers in CSF of iNPH Patients

iNPH is caused by a reduced absorption of CSF, which induces the accumulation of CSF in the ventricles. MRI imaging reveals enlarged ventricles with excess CSF, which compress the brain parenchyma, leading to dementia, gait disturbance, and urinary incontinence. Marker correlations were analyzed in CSF of iNPH patients, with scatter diagrams and Spearman’s rank correlation coefficients shown below for each marker combination (Figure 6, Table 5). Brain-derived markers were highly correlated with each other: Man-Tf vs. GlcNAc-Tf (rs = 0.82), Man-Tf vs. L-PGDS (rs = 0.75), and GlcNAc-Tf vs. L-PGDS (rs = 0.73). In contrast, Man-Tf showed a low correlation with Sia-Tf (rs = 0.37) and total protein (rs = 0.29). Nevertheless, Sia-Tf and total protein were well correlated with each other (rs = 0.70).

#### 2.4.2. Correlation of CSF Markers in Non-iNPH Patients

Marker correlations were analyzed in CSF samples from non-iNPH patients (Figure 7, Table 6). Brain-derived markers were significantly correlated with each other: Man-Tf vs. GlcNAc-Tf (rs = 0.48), Man-Tf vs. L-PGDS (rs = 0.42), and GlcNAc-Tf vs. L-PGDS (rs = 0.44). Man-Tf levels were not correlated with Sia-Tf (rs = 0.25) or total protein (rs = 0.25), whereas GlcNAc-Tf levels were correlated with Sia-Tf (rs = 0.45) and total protein (rs = 0.43). L-PGDS was correlated with Sia-Tf (rs = 0.47). Sia-Tf and total protein were well correlated with each other (rs = 0.83).

### 2.5. Diagnostic Accuracy of CSF Markers for SIH and iNPH

We next examined the diagnostic accuracy of CSF markers for SIH and iNPH, which showed significant differences in marker levels. The criteria for good markers are a sensitivity and specificity of more than 85%, together with an area under the ROC curve (AUC) greater than 0.9. SIH is mainly diagnosed on the basis of MRI imaging and the measurement of intracranial hypotension. As biochemical markers for SIH are barely established, CSF markers were examined for their diagnostic accuracy. Sensitivity values for Man-Tf, GlcNAc-Tf, L-PGDS, Sia-Tf, and the total protein levels were 97%, 77%, 77%, 46%, and 62%, respectively, while the specificity values were 95%, 75%, 75%, 95%, and 85%, respectively (Table 7). Man-Tf showed a 0.98 AUC, indicating that it would likely be a much better diagnostic marker than GlcNAc-Tf or L-PGDS.

Levels of GlcNAc-Tf and L-PGDS in CSF samples from iNPH patients were significantly lower than those of non-iNPH patients. CSF markers were examined for their diagnostic accuracy to differentiate iNPH from non-iNPH. The sensitivity values for Man-Tf, GlcNAc-Tf, L-PGDS, Sia-Tf, and the total protein levels were 62%,72%, 82%, 81%, and 46%, respectively, whereas their respective specificity values were 38%, 28%, 57%, 19%, and 54% (Table 8). All markers showed low specificities, indicating that they are not applicable for the diagnosis of iNPH.

### 2.6. CSF Markers after Shunt Operation

#### 2.6.1. Marker Levels at Post-Operative Days 1–6

Patients with iNPH usually undergo a “shunt operation”, in which excess CSF is bypassed from the ventricles to the peritoneal cavity. Following the operation, prior symptoms such as dementia and gait disturbance usually attenuate gradually (shunt responder), suggesting that the shunt system functions well and that CSF metabolism has normalized. Responders were examined for their marker levels with the expectation that the operation would change pre-operative marker levels. CSF specimens were withdrawn from the shunt valve over time post-operation, and marker levels were quantified and compared with pre-operative levels. Brain-derived markers slightly increased on post-operative days 1–3 and further increased on days 4–6 (Figure 8). Man-Tf, GlcNAc-Tf, and L-PGDS eventually showed 1.6-, 1.6- and 1.8-fold increases, respectively (Table 9). Blood-derived markers showed only subtle changes during the same period.

#### 2.6.2. Correlation Coefficients of CSF Marker Levels at Postoperative Days 1–6

The correlation coefficients for each marker combination were examined at post-operative days 1–6 (Table 10). Brain-derived markers were correlated with each other: Man-Tf vs. GlcNAc-Tf (rs = 0.72), Man-Tf vs. L-PGDS (rs = 0.66), and L-PGDS vs. GlcNAc-Tf (rs = 0.47). Other marker combinations showed low or no correlations.

#### 2.6.3. Marker Levels at Post-Operative Months 3, 6 and 12

CSF specimens were withdrawn at post-operative months 3, 6 and 12. The time course of the marker level changes was examined by comparing pre- and post-operative levels. The brain-derived marker levels increased significantly in the post-operative period, while the Sia-Tf and total protein levels increased only marginally (Figure 9, Table 11).

Correlation analyses revealed that the brain-derived markers correlated well with each other at each time point: Man-Tf vs. GlcNAc-Tf (rs = 0.59–0.76), Man-Tf vs. L-PGDS (rs = 0.64–0.74), and L-PGDS vs. GlcNAc-Tf (rs = 0.62–0.76) (Table 12). Blood-derived markers were correlated with Man-Tf at post-operative month 6 (rs = 0.51–0.54) and with GlcNAc-Tf at post-operative months 3 and 6 (rs = 0.58–0.76). Total protein was correlated with L-PGDS at post-operative month 3 (rs = 0.57), while Sia-Tf was correlated with the brain-derived markers at post-operative month 12 (rs = 0.42–0.55). These results suggest that the brain-derived markers were concomitantly increased and correlated with each other.

## 3. Discussion

We previously reported that the CSF levels of brain-derived glycoproteins, such as Man-Tf, GlcNAc-Tf, and L-PGDS, are well correlated with each other in neurodegenerative disorders, including AD [6]. The present results revealed that, except for L-PGDS in non-SIH, these CSF markers were significantly correlated in other neurological diseases such as depression, schizophrenia, SIH and iNPH. Although the latter two diseases showed different levels of markers from other diseases, the levels were still correlated.

Concerning the mechanisms that underlie the elevation of brain-derived proteins in SIH, it is known that the disease is caused by CSF leakage, which induces intracranial hypotension and subsequent brain dislocation, leading to orthostatic headache and dizziness, among other symptoms [16,17,19]. SIH is usually treated by the “blood patch” method, in which a patient’s own blood is injected into leakage sites, generating blood clots to stop the CSF leak. This therapy relieves the patient’s symptoms, suggesting that the leakage and loss of CSF are the primary causes of the disease for which a compensatory increase in CSF production is presumed. The results of the present study revealed that brain- and blood-derived protein levels were markedly elevated in SIH. Among the markers, L-PGDS is biosynthesized in the choroid plexus and then secreted into the CSF [25]. It is assumed that L-PGDS secretion in SIH increases concomitantly with enhanced CSF production. This might also be the case for GlcNAc-Tf, which is secreted from the choroid plexus [13,26]. This hypothesis may, at least partly, account for the high correlation of these markers in SIH. In addition, the two markers were correlated with Man-Tf, a neuron-derived protein [11]. One possible explanation is that Tf gene expression is controlled by common transcriptional regulation in neurons and choroid plexus epithelium. After the translation, Tf proteins are post-translationally modified with different glycans: mannose-terminated glycans in neurons and GlcNAc-terminated glycans in choroid plexus epithelium. Taken together, we consider that Tf isoforms and PGDF are markers of the enhancement of CSF production in SIH.

Next, in relation to the mechanisms associated with the elevation of blood-derived proteins in SIH, our previous study demonstrated that total protein, albumin, immunoglobulin, and Sia-Tf markedly increased in CSF, but not in serum [7]. The pattern of protein expression in the CSF was similar to that of serum, except for the presence of brain-derived proteins, suggesting that each blood protein was translocated from the blood into CSF with similar efficacies. The present results showed that the total protein and Sia-Tf levels significantly increased, and were correlated with each other in the CSF. It is also noteworthy that the levels of these markers were also highly correlated in other neurological diseases and controls (rs = 0.70~0.83). These results suggest that, regardless of marker levels, small amounts of blood proteins are translocated from the blood into the CSF with similar efficiency. It is also reported that, unlike blood capillaries that form the blood–brain barrier, choroid plexus capillaries are fenestrated and have no tight junctions, suggesting that blood proteins could diffuse into the CSF in a concentration dependent-manner at this site [27]. The level of diffusion may, at least partly, depend on the intracranial pressure (ICP) of the affected individual. As included in the diagnostic criteria, the ICP in SIH (0~60 mm H_2_O) is much lower than that of the controls (70–180 mm H_2_O) [17]. Consequently, a decreased ICP induces the dilatation of blood vessels and a concomitant increase in the intracranial blood volume, which might facilitate the diffusion of blood proteins into the CSF. We therefore assume that the elevation of blood-derived marker levels in the CSF of SIH patients is mainly due to the presence of intracranial hypotension.

Concerning the potential mechanisms that underlie marker changes in iNPH, it was previously demonstrated that affected patients show an accumulation of CSF in the ventricles, which is possibly due to a reduced absorption of CSF [28]. A secondary decline in CSF production is suspected. L-PGDS and GlcNAc-Tf levels in the CSF samples of iNPH patients were significantly lower than those of non-iNPH samples, but levels of other markers were not. In addition, the brain-derived marker levels were highly correlated with each other. L-PGDS and Man-Tf did not correlate with the blood-derived marker levels, whereas GlcNAc-Tf levels did correlate moderately. A shunt operation is the standard approach usually used to treat iNPH. In the present study, CSF marker levels in responders to the shunt operation were examined before and after patients underwent the procedure. In the shunt responders, the brain-derived marker levels in these patients tended to increase, even in post-operative days 1–6 (50~65%), compared to pre-operative levels, while the blood-derived markers scarcely increased (0~11%). The brain-derived marker levels were highly correlated with each other, but the blood-derived markers were not, suggesting that the former levels were controlled by a mechanism distinct from the latter, at least during this period. We also analyzed marker levels at post-operative months 3–12. The brain-derived marker levels were significantly higher than the pre-operative levels (49~113%), whereas the blood-derived markers increased only marginally (12~44%). The brain-derived markers were highly correlated with each other at all time points, with correlation coefficients at post-operative months 3, 6, and 12 found to be 0.59–0.64, 0.68–0.73, and 0.74–0.76, respectively. During this period, clinical symptoms were generally attenuated in responders, which suggests that CSF metabolism was enhanced in line with the increase in the markers. In contrast, blood-derived marker levels were only moderately correlated in this period. We therefore propose that brain-derived markers, but not blood-derived markers, could serve as indicators or biomarkers of altered CSF production.

Lastly, we discuss the diagnostic applications of the markers examined in our work. Mase et al., reported that L-PGDS levels were decreased in the CSF of NPH patients [29]. We also demonstrated a significant decrease in GlcNAc-Tf, in addition to L-PGDS. Given that L-PGDS showed a sensitivity of 82% and a specificity of 57%, while GlcNAc-Tf had a sensitivity of 96% and a specificity of 28%, our results suggest that these markers would not serve as good diagnostic markers for iNPH.

The present results revealed that all markers were elevated in SIH compared with the non-SIH CSF samples. Based on its sensitivity of 97%, specificity of 95%, and AUC of 0.98, our results suggest that Man-Tf could be a good diagnostic marker for SIH. In contrast, the other markers we examined would not be applicable for diagnostic purposes given their low sensitivity values (46~77%).

We previously demonstrated that Man-Tf derived from neurons could be used as a diagnostic marker for AD and MCI [11]. Indeed, Man-Tf levels in the CSF are highly correlated with those of phosphorylated-tau protein (p-tau), a marker for AD-specific neuronal death. A concomitant increase in Man-Tf and p-tau led the authors to develop an index, Man-Tf x p-tau, as a diagnostic marker. The index showed a sensitivity of 94% and a specificity of 89% for AD, and a sensitivity of 84% and a specificity of 90% for MCI. The present study revealed that Man-Tf was markedly increased in SIH, indicating that such an increase in Man-Tf is not valid as a specific marker for AD pathology. Nevertheless, measuring Man-Tf or the index would still be of diagnostic value for both diseases. SIH patients, who are usually less than 50 years old, mostly complain of orthostatic headaches, while AD and MCI patients, who are more likely to be in their sixties or older, tend to show signs of dementia. Together with the distinctive difference in symptoms and the peak age of onset, the Man-Tf measure should serve as a supportive diagnostic marker for AD and SIH. Tf captures ferric iron with very high affinity (Kd = 1 × 10^−20^), which prevents Fe-dependent cellular toxicity. Indeed, Fe toxicity is suspected to induce neuronal death in AD pathology. Man-Tf, mainly produced by neurons, may play a role as a local protein that protects against iron toxicity, although the mechanisms underlying this have yet to be elucidated.

## 4. Materials and Methods

### 4.1. Patients and CSF Samples

Patients were consecutively recruited from Fukushima Medical University (FMU), Juntendo University (JTN), Sanno Hospital (SNO), and the National Center for Neuropsychiatric Disease (NCNP). The number of patients, age (mean ± S.D.), male/female distribution, and hospitals for CSF collection for each disease type are listed in Table 13. Depression and schizophrenia were diagnosed based on “Diagnostic and Statistical Manual of Mental Disorders-*DSM-5*”. SIH was suspected by clinical presentation, particularly by orthostatic headaches, MRI and/or computed tomography (CT), and RI scintigraphy. Patients were diagnosed based on the International Classification of Headache Disorders, 3rd edition (beta-version) [18] and the diagnostic criteria by Schievink et al. [17]: (i) morphological evidence of CSF leakage, such as pachymeningeal enhancement, on cranial MRI and/or low CSF opening pressure (≤60 mm H_2_O), (ii) no recent history of dural puncture, and (iii) not attributable to another disorder. The iNPH was diagnosed according to clinical guidelines issued by the Japanese Society of Normal Pressure Hydrocephalus, Version 3 [30]. Briefly, iNPH was suspected based on clinical symptoms such as gait disturbance, dementia, and urinary incontinence. In addition, excess CSF in the ventricles (ventriculomegaly) was observed by CT or MRI. Patients with suspected iNPH underwent a tap test, which is a time-specific temporal removal of 30 mL of CSF by lumbar puncture. When the patients showed only a subtle improvement in their symptoms after CSF removal (tap test-negative), they were classified into the non-iNPH group. When the patients showed an evident improvement in their symptoms, they were classified into the probable iNPH group and underwent a shunt operation that bypassed excess CSF into the peritoneal cavity. Either a lumbar–peritoneal (LP) or a ventricle–peritoneal (VP) shunt was performed by using a Codman Hakim programmable valve with Siphonguard (#6244000, Integra Japan Co., Ltd., Tokyo, Japan) or a Medtronic Strata system (Medtronic Japan Co., Ltd., Tokyo, Japan). Patients showing an improvement in their symptoms after surgery were diagnosed as responders or definite iNPH. Several responders underwent a time-course analysis after the shunt operation, in which specimens were withdrawn from the shunt valve. The iNPH condition is occasionally a comorbidity associated with AD. Comorbid patients were excluded based on elevated levels (>50 pg/mL) of p-tau protein, a specific marker of neuronal death in AD pathophysiology.

### 4.2. Quantificatuin of CSF Proteins

CSF samples were aliquoted and stored at −80 °C until use. Repeated freeze and thawing (>two times) was avoided. Each assay was performed in triplicate. Man-Tf, GlcNAc-Tf, and Sia-Tf were quantified in lectin-ELISAs according to our previous studies [11,31]. Lectin probes for detecting Man-Tf, Sia-Tf, and GlcNAc-Tf were rBC2L-A (#026-18691) and SSA (#197-1037) from FUJIFILM Wako, Tokyo, Japan, and rPVL from Medical and Biological Laboratories Co., Ltd., Nagoya, Japan, respectively. Prior to use, lectins were biotinylated using Ez-Link NHS-Biotin (#20217, Thermo Fisher Scientific, Waltham, MA, USA). The assay method for detecting Man-Tf and Sia-Tf was as follows. Briefly, 96-well plates (C8 Maxisorp Nunc-Immuno Module plate, Nunc, Roskilde, Denmark) were coated with rabbit anti-Tf antibody (1 μg/mL) (A0061, Dako Ltd./Agilent Technologies, Inc., Santa Clara, CA, USA) in 100 mM of carbonate buffer at 4 °C overnight. Plates were washed with Tris-buffered saline (TBS) and then blocked at room temperature for 1 h with 10% N101 (S410-0301, NOF Corp., Tokyo, Japan) in TBS. The plate was washed with TBS containing 0.05% Tween 20 (#1706531, Bio-Rad Laboratories, Inc., Hercules, CA, USA) (TBST). For quantifying Man-Tf and GlcNAc-Tf, the CSF samples were pre-treated at 55 ℃ for 1 h in the presence of 10–20 μL of phosphate-buffered saline (PBS), containing 0.6% 2-mercaptoethanol (#1610710, Bio-Rad Laboratories, Inc.) and 0.003% SDS (S0588, Tokyo Chemical Industry Co., Ltd., Tokyo, Japan). The heat treatment was unnecessary for detecting Sia-Tf. The sample solution was appropriately diluted with TBST, applied to a plate, and incubated overnight at 4 °C. The plate was then washed three times with TBST. For quantifying Man-Tf, plates were incubated with biotinylated-rBC2L-A lectin (1 μg/mL) in TBST containing 10 mM CaCl_2_ (TBST-CaCl_2_) at room temperature for 2 h. Plates were washed twice with TBST-CaCl_2_, and horseradish peroxidase-labeled streptavidin (50 ng/mL) (N100, Thermo Fisher Scientific) in TBST-CaCl_2_ was added and incubated for 2 h at room temperature. The plates were washed twice with TBST and then a TMB Microwell Peroxidase Substrate System (#50-76-11, Kirkegaard and Perry Laboratories, Inc., Gaithersburg, MD, USA) was added. Color development was stopped by adding 1N phosphoric acid. Absorbances were measured at 450 nm by a Varioskan LUX multimode microplate reader (Thermo Fisher Scientific). Assays were performed in triplicate.

Recombinant BC2L-A lectin requires 10 mM CaCl_2_ for maximum binding, but other lectins do not. For quantifying Sia-Tf, assays similar to the rBC2L-A lectin-ELISA described above were performed, except that, instead of biotinylated rBC2L-A, biotinylated SSA lectin (1 μg/mL) was used with TBST buffer without CaCl_2_. For quantifying GlcNAc-Tf, rPVL lectin was immobilized, and captured GlcNAc-Tf was detected with anti-Tf antibody (0.2 μg/mL).

Total Tf was quantified with a Human Transferrin ELISA Quantitation Set (E80-128-23, Bethyl Laboratories, Inc., Montgomery, TX, USA). The protein concentration was estimated using a Micro-BCA protein assay kit (23235, Thermo Fisher Scientific-JP, Tokyo, Japan).

### 4.3. Statistical Analyses

Statistical analyses were performed using SPSS software (version 26). Data normality was examined using the Shapiro–Wilk test. Spearman’s rank correlation coefficients (rs) and Pearson’s correlation coefficients (r) were analyzed with and without data normality, respectively. Significant differences among multiple comparisons were assessed by the Kruskal–Wallis method followed by Bonferroni correction. Non-parametric comparisons were assessed by a Mann–Whitney U test.

## 5. Conclusions

The present study reveals that CSF contains brain-derived glycoproteins such as L-PGDS, Man-Tf, and GlcNAc-Tf, and that their CSF levels were highly correlated with each other in several neurological diseases. INPH, disorders featuring abnormal CSF metabolism, was treated with shunt operation. After the surgical intervention, brain-derived glycoproteins increased over a year and their CSF levels were highly correlated with each other. The present observational and interventional study reveals that the glycoproteins are novel biomarkers for CSF metabolism.

## Figures and Tables

**Figure 1 ijms-24-06084-f001:**
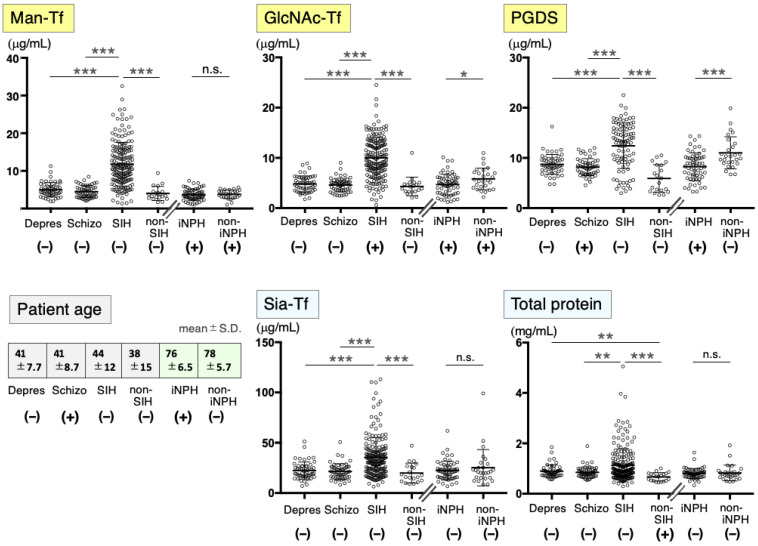
CSF marker levels in neurological diseases. CSF marker levels were analyzed in depression, schizophrenia, SIH, non-SIH, iNPH, and non-iNPH patients. Tf isoforms such as Man-Tf, GlcNAc-Tf, and Sia-Tf were quantified by a lectin/antibody-sandwich ELISA; the glycan-specific probes used were rBC2L-A, rPVL, SSA lectin, respectively. L-PGDS levels were quantified with a sandwich ELISA kit. Total protein was quantified by the micro-BCA method. Ages of patients in each disease are listed in boxes (mean ± S.D.). Data normality for each marker is indicated with a plus (+) or minus (−) sign. Multiple comparisons among depression, schizophrenia, SIH, and non-SIH patients were assessed by the Kruskal–Wallis method, followed by Bonferroni correction. Comparisons of iNPH and non-iNPH were assessed by a Mann–Whitney U test. All combinations of clinical groups were assessed, and combinations showing significant differences are indicated with horizontal bars. Probability values: * *p* < 0.05, ** *p* < 0.01, *** *p* < 0.001. n.s.: not significant.

**Figure 2 ijms-24-06084-f002:**
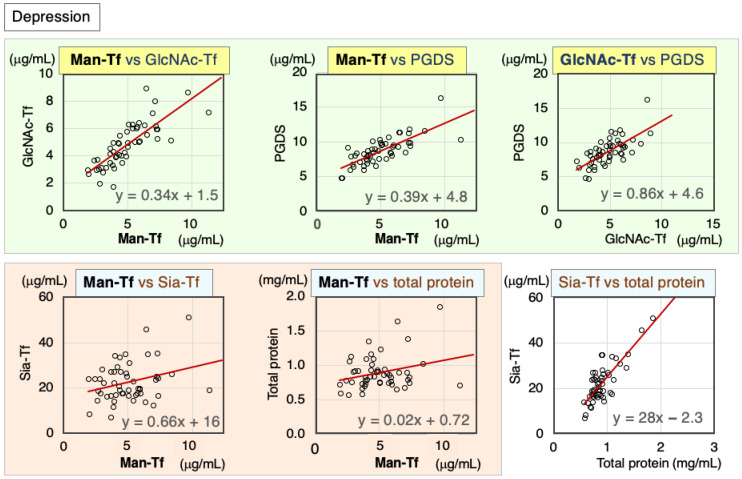
Scatter diagrams and regression lines of marker combinations in the CSF of depression patients.

**Figure 3 ijms-24-06084-f003:**
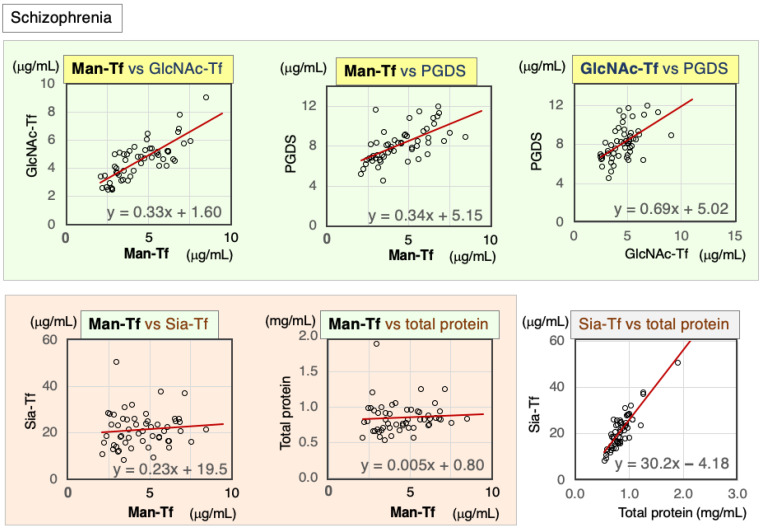
Scatter diagrams and regression lines of marker combinations in the CSF of schizophrenia patients.

**Figure 4 ijms-24-06084-f004:**
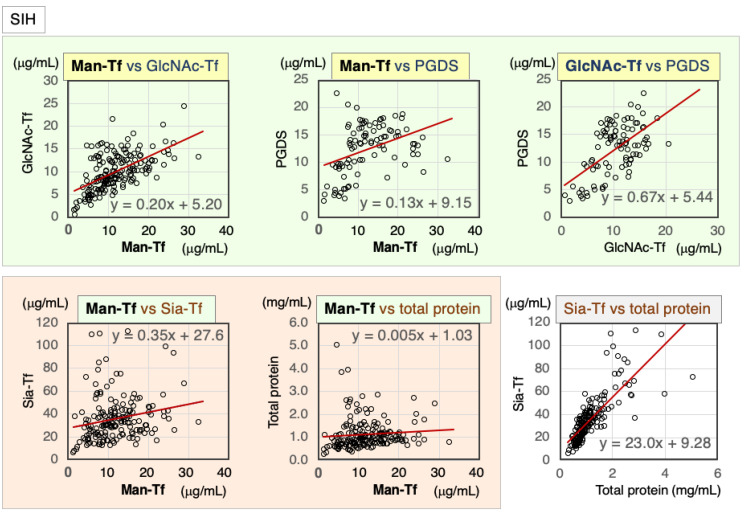
Scatter diagrams and regression lines for marker combinations in SIH patients.

**Figure 5 ijms-24-06084-f005:**
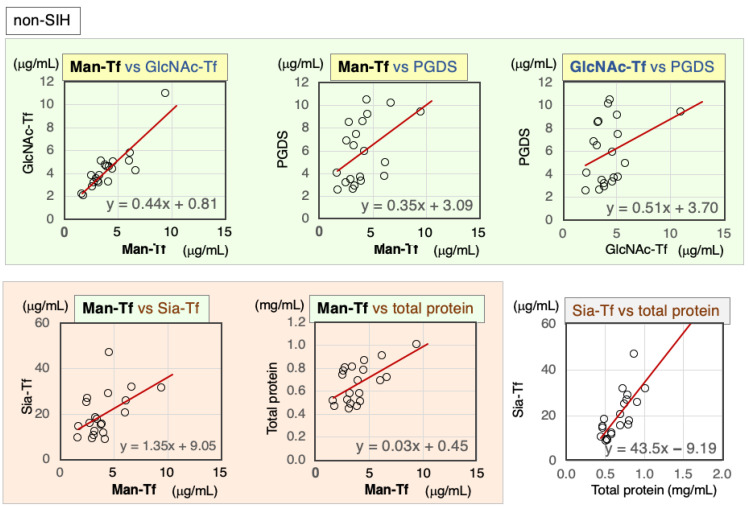
Scatter diagrams and regression lines for marker combinations in non-SIH patients.

**Figure 6 ijms-24-06084-f006:**
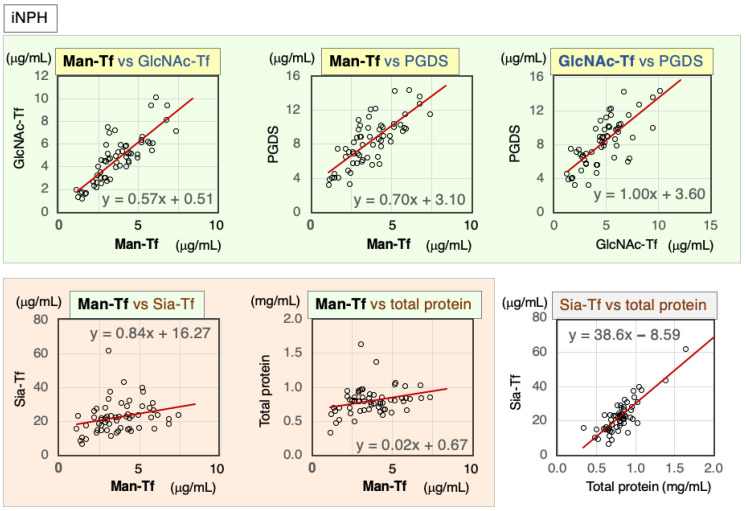
Scatter diagrams and regression lines of marker combinations for iNPH patients.

**Figure 7 ijms-24-06084-f007:**
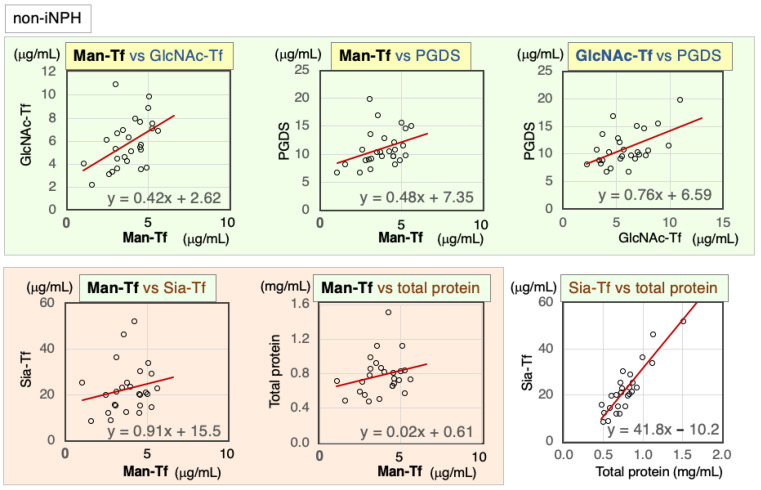
Scatter diagrams and regression lines for marker combinations in non-iNPH patients.

**Figure 8 ijms-24-06084-f008:**
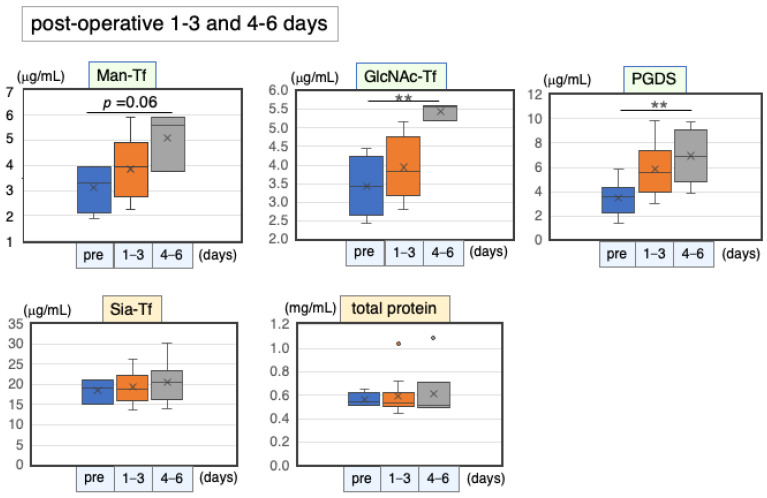
CSF marker levels at post-operative days 1–3 and 4–6. Horizontal lines within boxes show median values; boxes exclude upper and lower interquartile ranges; whiskers indicate maximum and minimum values. Outliers are indicated by dots. Probability level: ** *p* < 0.01.

**Figure 9 ijms-24-06084-f009:**
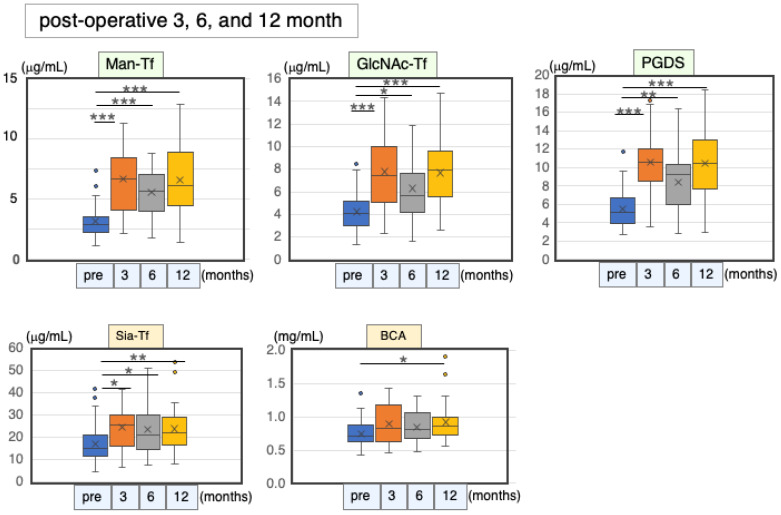
CSF marker levels at post-operative months 3, 6, and 12. Horizontal lines within boxes show median values; boxes exclude upper and lower interquartile ranges; whiskers indicate maximum and minimum values. Outliers are indicated by dots. Probability level: * *p* < 0.05, ** *p* < 0.01, *** *p* < 0.001.

**Table 1 ijms-24-06084-t001:** Correlation coefficients of markers in depression patients.

Depression	Nor.	Correl. Coeff.	Man-Tf	GlcNAc-Tf	L-PGDS	Sia-Tf	Total Protein
**Man-Tf**	−	rs	1.00	0.87 **	0.77 **	0.20	0.16
**GlcNAc-Tf**	+	rs	0.87 **	1.00	0.70 **	0.19	0.13
**L-PGDS**	−	rs	0.77 **	0.70 **	1.00	0.44 **	0.33 *
**Sia-Tf**	−	rs	0.20	0.19	0.44 **	1.00	0.65 **
**Total protein**	−	rs	0.16	0.13	0.33 *	0.65 **	1.00

Spearman’s rank correlation coefficients (correl. Coeff.) among Man-Tf, GlcNAc-Tf, and L-PGDS are highlighted in yellow, while that of Sia-Tf vs. total protein is highlighted in pink. Data normality (Nor.) is indicated by a plus (+) or minus (−) sign. Probability level: * *p* < 0.05, ** *p* < 0.01.

**Table 2 ijms-24-06084-t002:** Correlation coefficients of markers in CSF of schizophrenia patients.

Schizophrenia	Nor.	Correl. Coeff.	Man-Tf	GlcNAc-Tf	L-PGDS	Sia-Tf	Total Protein
**Man-Tf**	−	rs	1.00	0.76 **	0.68 **	0.13	0.18
**GlcNAc-Tf**	−	rs	0.76 **	1.00	0.55 **	0.00	0.00
**L-PGDS**	+	rs	0.68 **	0.55 **	1.00	0.49 **	0.46 **
**Sia-Tf**	−	rs	0.13	0.00	0.49 **	1.00	0.79 **
**Total protein**	−	rs	0.18	0.00	0.46 **	0.79 **	1.00

Spearman’s rank correlation coefficients (correl. coeff.) among Man-Tf, GlcNAc-Tf, and L-PGDS are highlighted in yellow, while that of Sia-Tf vs. total protein is highlighted in pink. Data normality (Nor.) is shown by a plus (+) or minus (−) sign. Probability level: ** *p* < 0.01.

**Table 3 ijms-24-06084-t003:** Correlation coefficients and data normality of markers.

SIH	Nor.	Correl. Coeff.	Man-Tf	GlcNAc-Tf	L-PGDS	Sia-Tf	Total Protein
**Man-Tf**	−	rs	1.00	0.57 **	0.45 **	0.26 **	0.27 **
**GlcNAc-Tf**	+	rs	0.57 **	1.00	0.47 **	0.70 **	0.75 **
**L-PGDS**	−	rs	0.45 **	0.47 **	1.00	0.53 **	0.57 **
**Sia-Tf**	−	rs	0.26 **	0.70 **	0.53 **	1.00	0.86 **
**Total protein**	−	rs	0.27 **	0.75 **	0.57 **	0.86 **	1.00

Spearman’s rank correlation coefficients (correl. coeff.) among Man-Tf, GlcNAc-Tf, and L-PGDS are highlighted in yellow, while that of Sia-Tf vs. total protein is highlighted in pink. Correlation coefficients between blood-derived markers and GlcNAc-Tf or L-PGDS are highlighted in blue. Data normality (Nor.) is shown by a plus (+) or minus (−) sign. Probability level: ** *p* < 0.01.

**Table 4 ijms-24-06084-t004:** Correlation coefficients and data normality of markers.

Non-SIH	Nor.	Correl. Coeff.	Man-Tf	GlcNAc-Tf	L-PGDS	Sia-Tf	Total Protein
**Man-Tf**	−	rs	1.00	0.79 **	0.55 **	0.51 *	0.44 *
**GlcNAc-Tf**	−	rs	0.79 **	1.00	0.27	0.46 *	0.50 *
**L-PGDS**	−	rs	0.55 *	0.27	1.00	0.51 *	0.67 **
**Sia-Tf**	+	rs	0.51 *	0.46 *	0.51 *	1.00	0.73 **
**Total protein**	−	rs	0.44 *	0.50 *	0.67 **	0.73 **	1.00

Spearman’s rank correlation coefficients (correl. coeff.) among Man-Tf, GlcNAc-Tf, and L-PGDS are highlighted in yellow, while that of Sia-Tf vs. total protein is highlighted in pink. Correlation coefficients between brain-derived and blood-derived markers are highlighted in blue. Data normality (Nor.) is shown by a plus (+) or minus (−) sign. Probability level: * *p* < 0.05, ** *p* < 0.01.

**Table 5 ijms-24-06084-t005:** Correlation coefficients and data normality of markers.

iNPH	Nor.	Correl. Coeff.	Man-Tf	GlcNAc-Tf	L-PGDS	Sia-Tf	Total Protein
**Man-Tf**	+	rs	1.00	0.82 **	0.75 **	0.37 **	0.29 *
**GlcNAc-Tf**	+	rs	0.82 **	1.00	0.73 **	0.60 **	0.56 **
**L-PGDS**	+	rs	0.75 **	0.73 **	1.00	0.46 **	0.53 **
**Sia-Tf**	−	rs	0.37 **	0.60 **	0.46 **	1.00	0.70 **
**Total protein**	−	rs	0.29 *	0.56 **	0.53 **	0.70 **	1.00

Spearman’s rank correlation coefficients (correl. coeff.) among Man-Tf, GlcNAc-Tf, and L-PGDS are highlighted in yellow, while that of Sia-Tf vs. total protein is highlighted in pink. Correlation coefficients between brain-derived and blood-derived markers are highlighted in blue. Data normality (Nor.) is shown by a plus (+) or minus (−) sign. Probability level: * *p* < 0.05, ** *p* < 0.01.

**Table 6 ijms-24-06084-t006:** Correlation coefficients and data normality of markers.

Non-iNPH	Nor.	Correl. Coeff.	Man-Tf	GlcNAc-Tf	L-PGDS	Sia-Tf	Total Protein
**Man-Tf**	+	rs	1.00	0.48 *	0.42 **	0.25	0.25
**GlcNAc-Tf**	+	rs	0.48 *	1.00	0.44 *	0.45 *	0.43 *
**L-PGDS**	−	rs	0.42 *	0.44 *	1.00	0.28	0.47 *
**Sia-Tf**	−	rs	0.25	0.45 *	0.28	1.00	0.83 **
**Total protein**	−	rs	0.25	0.43 *	0.47 *	0.83 **	1.00

Spearman’s rank correlation coefficients (correl. coeff.) among Man-Tf, GlcNAc-Tf, and L-PGDS are highlighted in yellow, while that of Sia-Tf vs. total protein is highlighted in pink. Correlation coefficients between GlcNAc-Tf and blood-derived markers, and between L-PGDS and total protein are highlighted in blue. Data normality (Nor.) is indicated by a plus (+) or minus (−) sign. Probability level: * *p* < 0.05, ** *p* < 0.01.

**Table 7 ijms-24-06084-t007:** Diagnostic accuracy for differentiating SIH and non-SIH in patients.

Non-SIH vs. SIH	Cutoff (μg/mL)	AUC	Sensitivity (%)	Specificity (%)
**Man-Tf**	6.4	0.98	97	95
**GlcNA-cTf**	9.2	0.59	77	75
**L-PGDS**	9.1	0.78	77	75
**Sia-Tf**	31.8	0.76	46	95
**Total protein**	810	0.77	62	85

Sensitivity and specificity values greater than 85% are highlighted in yellow.

**Table 8 ijms-24-06084-t008:** Diagnostic accuracy for differentiating iNPH from non-iNPH.

Non-iNPH vs. iNPH	Cutoff (μg/mL)	AUC	Sensitivity (%)	Specificity (%)
**Man-Tf**	6.0	0.55	62	38
**GlcNAc-Tf**	3.1	0.65	72	28
**L-PGDS**	8.9	0.74	82	57
**Sia-Tf**	12.7	0.52	81	19
**Total protein**	750	0.55	46	54

**Table 9 ijms-24-06084-t009:** CSF marker levels at post-operative days 1–3 and 4–6.

	Pre-Operation		1–3 Days			4–6 Days		
	n	(μg/mL)	n	(μg/mL)	%	n	(μg/mL)	%
**Man-Tf**	4	6.2 ± 2.0 *	12	7.7 ± 2.3 *	124	3	10.2 ± 2.4 *	163
**GlcNAc-Tf**	4	3.4 ± 0.8	12	3.9 ± 0.8	115	3	5.5 ± 0.2	159
**L-PGDS**	4	3.9 ± 1.4	12	5.8 ± 2.3	151	8	7.0 ± 2.3	179
**Sia-Tf**	3	18.5 ± 3.0	12	19.4 ± 3.7	105	6	20.6 ± 5.5	111
**Total protein**	4	563 ± 60	12	593 ± 158	105	6	616 ± 233	109

* mean ± S.D. n: number of patients analyzed. Marker increase (%) greater than 115% are highlighted in yellow.

**Table 10 ijms-24-06084-t010:** Correlation coefficients and data normality of markers at post-operative days 1–6.

Post-Operative 1–6 Days	Nor.	Correl. Coeff.	Man-Tf	GlcNAc-Tf	L-PGDS	Sia-Tf	Total Protein
**Man-Tf**	+	rs	1.00	0.72 **	0.69 **	0.18	−0.19
**GlcNAc-Tf**	+	rs	0.72 **	1.00	0.71 **	0.40	0.21
**L-PGDS**	+	rs	0.69 **	0.71 **	1.00	0.24	−0.18
**Sia-Tf**	+	rs	0.18	0.40	0.24	1.00	0.26
**Total protein**	−	rs	−0.19	0.21	−0.18	0.26	1.00

Spearman’s rank correlation coefficients (correl. coeff.) among Man-Tf, GlcNAc-Tf, and L-PGDS are highlighted in yellow, while that of Sia-Tf vs. total protein is highlighted in pink. Data normality (Nor.) is indicated by a plus (+) or minus (−) sign. Probability level: ** *p* < 0.01.

**Table 11 ijms-24-06084-t011:** CSF marker levels at post-operative months 3, 6, and 12.

	Pre-Operation		3 Months			6 Months			12 Months		
	n	(μg/mL)	n	(μg/mL)	%	n	(μg/mL)	%	n	(μg/mL)	%
**Man-Tf**	48	6.3 ± 2.5 *	16	13.4 ± 5.4 *	213	26	11.1 ± 4.0 *	176	34	13.2 ± 5.4 *	209
**GlcNAc-Tf**	48	4.2 ± 1.7	16	7.8 ± 3.4	184	26	6.3 ± 2.8	149	34	7.7 ± 3.1	182
**L-PGDS**	44	5.5 ± 1.9	12	10.6 ± 3.8	193	21	8.4 ± 3.1	154	30	10.5 ± 3.6	192
**Sia-Tf**	48	17.0 ± 7.5	16	24.5 ± 9.5	144	26	23.6 ± 11.8	139	34	23.8 ± 9.8	140
**Total protein**	44	750 ± 187	13	892 ± 313	119	23	840 ± 223	112	31	921 ± 284	123

* mean ± S.D. n: number of patients analyzed. Marker increase (%) greater than 149% are highlighted in yellow.

**Table 12 ijms-24-06084-t012:** Correlation coefficients and data normality of markers at post-operative months 3, 6, and 12.

3 Months		Man-Tf	GlcNAc-Tf	L-PGDS	Sia-Tf	Total Protein
Man-Tf	rs	1.00	0.59 *	0.64 *	0.43	0.29
GlcNAc-Tf	rs	0.59 *	1.00	0.62 *	0.58 *	0.76 **
L-PGDS	rs	0.64 *	0.62 *	1.00	0.32	0.57 *
Sia-Tf	rs	0.43	0.58 *	0.32	1.00	0.64 *
Total protein	rs	0.29	0.76 **	0.57 *	0.64 *	1.00
** 6 months **		** Man-Tf **	** GlcNAc-Tf **	** L-PGDS **	** Sia-Tf **	** Total Pprotein **
Man-Tf	r	1.00	0.73 **	0.68 **	0.51 **	0.54 **
GlcNAc-Tf	r	0.73 **	1.00	0.70 **	0.58 **	0.64 **
L-PGDS	r	0.68 **	0.70 **	1.00	0.42	0.39
Sia-Tf	r	0.51 **	0.58 **	0.42	1.00	0.55 **
Total protein	r	0.54 **	0.64 **	0.39	0.55 **	1.00
** 12 months **		** Man-Tf **	** GlcNAc-Tf **	** L-PGDS **	** Sia-Tf **	** Total Pprotein **
Man-Tf	rs	1.00	0.76 **	0.74 **	0.42 *	0.23
GlcNAc-Tf	rs	0.76 **	1.00	0.76 **	0.55 **	0.31
L-PGDS	rs	0.74 **	0.76 **	1.00	0.48 **	0.26
Sia-Tf	rs	0.42 *	0.55 **	0.48 **	1.00	0.44 *
Total protein	rs	0.23	0.31	0.26	0.44 *	1.00

Spearman’s rank correlation coefficients (rs) for markers at 3 and 12 months and Pearson’s correlation coefficients (r) at 6 months are indicated. Coefficients among Man-Tf, GlcNAc-Tf, and L-PGDS are highlighted in yellow, those of GlcNAc-Tf vs. blood-derived proteins are highlighted in blue, and those of Sia-Tf and total proteins *p* < 0.05 are highlighted in pink. Data normality (Nor.) is indicated by a plus (+) or minus (−) sign. Probability level: * *p* < 0.05, ** *p* < 0.01.

**Table 13 ijms-24-06084-t013:** Patient Characteristics.

Disease	Age * (Years)	Patient Number	Gender (M/F)	Hospital
Depression	41 ± 7.7	52	24/28	NCNP
Schizophrenia	41 ± 8.7	54	27/27	NCNP
SIH: spontaneous intracranial hypotension	44 ± 12	199	85/114	SNO, MTK
Non-SIH **	33 ± 11	20	7/13	SNO, FMU
iNPH: idiopathic normal pressure hydrocephalus	76 ± 6.5	60	36/24	JTN, FMU
iNPH for time course study	73 ± 5.7	48	35/13	JTN, FMU
Non-iNPH	78 ± 5.7	27	15/12	JTN, FMU

* mean ± S.D., ** Cervical injury with or without car accident (ICP > 60 mm H_2_O).

## Data Availability

All data are contained within the article.

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
