# Peer review of "Brain-Derived Major Glycoproteins Are Possible Biomarkers for Altered Metabolism of Cerebrospinal Fluid in Neurological Diseases"

_ijms, 2023, doi:10.3390/ijms24076084_

Round 1

Reviewer 1 Report

This study is a continuation of previous publications by Hoshi and his friends. (reference 6). The study aims to be used as a biomarker by measuring the levels of brain-derivated glycoproteins (especially GlcNAc-Tf and Man-Tf), in neuropsychiatric diseases such as depression, schizophrenia, neurodegenerative diseases, and cerebrospinal fluid circulation disorders. This study is valuable study because it aims to develop new biomarkers and use them in clinical practice. I think the method of study is appropriate, as a Neurologist. As a result of this study and previous study, Man-TF levels were found to be high in both Alzheimer's disease and spontaneous intracranial hypotension. Researchers suggest Man-TF as a supporting biomarker for Alzheimer's and spontaneous intracranial hypotension. Alzheimer's diease and spontaneous intracranial hypotension are very different diseases as a patient age group, clinical characteristics, and pathogenesis. Therefore, in Alzheimer's disease, the Man-TF can be used in clinical practice as a biomarker to support the diagnosis. To increase the value of the work, Man-Tf levels must be measured and compared in patients with non-Alzheimer dementia. I think there is no problem with the reference, table, and figures in the study. I think this study is suitable for publication.

Author Response

Thanks for encouraging comments. Reviewer #2, #3 and #4 also recommended extensive editing of English language and style. We modified the manuscript to improve English quality and correct mistakes.  In addition, we are planning to ask English editing by IJMS editorial system.

Reviewer 2 Report

Hoshi et al, in this research study aimed to examine the biomarkers and their correlations in neurological disorders. First they measured levels of several biomarkers in CSF of patients having neurological disorders. They continued to show it in psychiatric disease patients depression patients, schizophrenia patients, CSF markers in SIH and so on. They observed that marker levels were correlated each other at the post-operative period. They conclude that brain-derived markers are well correlated in CSF of various neurological diseases and their correlations are kept even after the surgical intervention. Further, brain-derived proteins could be biomarkers.

Overall the article is interesting and the study is well organized. I would recommend it for publication after Language editing. Many sentences need were difficult to understand. Please proof read the whole article before submission.

Author Response

Language editing: Reviewer #1, #3 and #4 recommended extensive editing of English language and style. We modified the manuscript to improve English quality and correct mistakes.  In addition, we are planning to ask English editing by IJMS editorial system.

Introduction: General backgrounds on CSF metabolism might insufficient. Reviewer #3, and #4 also recommended providing sufficient background in introduction section. Accordingly, we inserted sentences in introduction section: line 61~73, line 83~88, line 108~114.

Reviewer 3 Report

The present paper investigates CSF levels of Man-Tf, GlcNAc-Tf, PGDS, Sia-Tf and total CSF protein in a large cohort of patients with depression, schizophrenia, SIH, non-SIH, iNPH and non-iNPH. The main outcome is that SIH patients showed very high and diverse levels of all markers, comparing with non-SIH, indicating that both brain- and blood-derived proteins are markedly increased in SIH. Likewise, in iNPH, there was an increase in GlcNAc-Tf and PGDS levels compared to non-iNPH.

Major comments:

1. The study design could be improved. The authors include completely heterogenous study groups, i.e. two psychiatric disorders (depression and schizophrenia) and two disorders of CSF circulation (SIH and iNPH). The rationale behind this study design should be explained. It would seem more reasonable to present data on these two groups in two different papers.

2. Control subjects are lacking. In the absence of a control group, conclusions regarding CSF biomarker levels are difficult to be drawn.

3. The clinical usefulness of the increase of all proteins in SIH vs. nonSIH is questionable. SIH is diagnosed with measurement of CSF pressure after lumbar puncture. It thus seems redundant to look into alterations in CSF proteins for this diagnosis.

4. The group non SIH is not described in the Methods section. This is important to be clarified, since the main conclusions regard the SIH vs. nonSIH differentiation.

5. The authors should discuss the possible role of raustrocaudal CSF protein gradient in SIH. Since these patients have a CSF leak, CSF from lumbar puncture may in fact result in sampling from a more rostral point, which may in part explain the increase in all CSF proteins.

6. The distinction between NPH and nonNPH based on response is problematic, considering that many NPH patients may not improve after ventriculo-peritoneal shunting when this is performed late in the disease course. It would be interesting however to look into differences in CSF proteins in differentiating responders vs. non-responders in the NPH cohort.

7. The Introduction could be improved by including data relating to the biological significance of the CSF proteins measured in this paper.

8. The discussion section could be improved by analyzing the biological implications of the study findings

Author Response

Major comments:

Comment #1

The study design could be improved. The authors include completely heterogenous study groups, i.e. two psychiatric disorders (depression and schizophrenia) and two disorders of CSF circulation (SIH and iNPH). The rationale behind this study design should be explained. It would seem more reasonable to present data on these two groups in two different papers.

Response to Comment #1

Aim of this study is testing the hypothesis that the brain-derived proteins are correlated in CSF of various neurological diseases. The study design included observational and interventional study. In observational study, the markers need to be analyzed as many diseases as possible, which include heterogenous study groups with different pathophysiology. The marker correlations were shown in two psychiatric disorders (depression and schizophrenia) and two disorders of CSF circulation (SIH and iNPH) in the present study while neurodegenerative disorders in the previous study. These observational studies indicate that, regardless of the patient’s pathophysiology, the markers were correlated each other. The postoperative analysis for iNPH patients (shunt responders) was the interventional study. After shunt operation, the markers were increased and still correlated well at any time point, suggesting that the marker levels are controlled by similar mechanisms even after surgical intervention.

It is possible to divide the present data into two different papers, but we prefer to publish the data in the single paper, which includes observational and interventional study simultaneously.

Comment #2

Control subjects are lacking. In the absence of a control group, conclusions regarding CSF biomarker levels are difficult to be drawn.

Response to Comment #2

I agree with the reviewer that control is important. The best control specimens are CSF obtained from age-matched healthy subjects. Our ethical committee, however, do not allow us to do lumbar puncture for healthy subject due to its invasiveness. Therefore, non-iNPH is used as controls, who was suspected to be iNPH but their Evans index is less than 0.3 and/or tap test is negative. Non-iNPH and iNPH were defined based on “guidelines for management of idiopathic normal pressure hydrocephalus: second edition” [reference No.27]. Another control is non-SIH subject, which is described in “Response to comment #4” and patients’ information is inserted in Table 13.

Comment #3

The clinical usefulness of the increase of all proteins in SIH vs. non-SIH is questionable. SIH is diagnosed with measurement of CSF pressure after lumbar puncture. It thus seems redundant to look into alterations in CSF proteins for this diagnosis.

Response to comment #3

SIH was diagnosed based on International Classification of Headache Disorders, 3rd edition (beta version) [reference No.16] and the diagnostic criteria reported by Schievink et al. [reference No.15] as follows.

(1) Morphological evidence of CSF leakage such as pachymeningeal enhancement on cranial MRI and/or low CSF opening pressure (≤60mm H2O).

(2) No recent history of dural puncture.

(3) Not attributable to another disorder.

As pointed out by the reviewer, CSF pressure is the most important criterion. Diagnostic criteria, however, include morphological changes evidenced by MRI or CT, indicating that multiple tests are preferable for confirming the diagnosis. So far biochemical markers for SIH has never been available. We believe that quantifying brain-derived proteins could be quantitative biomarkers for supporting SIH diagnosis.

Comment #4

The group non SIH is not described in the Methods section. This is important to be clarified, since the main conclusions regard the SIH vs. nonSIH differentiation.

Response to comment #4

I agree with the reviewer that I should describe non-SIH in detail. Non-SIH had histories of cervical injury: nine out of 20 subjects had car accidents while 11 did not had. They had a headache, but their intracranial pressure was more than 60mm H2O and no morphological change was observed on MRI or CT. Accordingly, we inserted the patient information in Table 13.

Comment #5

The authors should discuss the possible role of raustrocaudal CSF protein gradient in SIH. Since these patients have a CSF leak, CSF from lumbar puncture may in fact result in sampling from a more rostral point, which may in part explain the increase in all CSF proteins.

Response to comment #5

The reviewer’s comment is important. Based on the assumption by the reviewer, we divided SIH patients into three groups. Patients showing CSF leakages at the cervical and/or thoracic portions, which are more rostral than lumbar puncture (C/Th group). Patients showing CSF leakages at lumbar portions, which are more caudal (L group). Patients showing CSF leakages at multiple sites including lumbar portions were eliminated from the analysis because we did not assume a responsible portion for marker changes. Marker levels were compared in C/Th and L groups. Man-Tf levels in C/Th and L groups were 9.9+5.3 and 7.2+5.5, respectively, and difference was not significant (p = 0.08). GlcNAc-Tf levels in C/Th and L groups were 8.3+4.3 and 6.8+4.6, respectively, and difference was not significant (p = 0.13). PGDS levels in C/Th and L groups were 13.0+4.6 and 9.7+5.0, respectively, and difference was not significant (p = 0.17). The result suggests that raustrocaudal CSF protein gradient in SIH has a little effect on marker elevation, if any.

Comment #6

The distinction between NPH and non-NPH based on response is problematic, considering that many NPH patients may not improve after ventriculo-peritoneal shunting when this is performed late in the disease course. It would be interesting however to look into differences in CSF proteins in differentiating responders vs. non-responders in the NPH cohort.

Response to comment #6

For interventional study, it is essential that intervention itself should function well. For example, shunt responders showed improvement of symptoms, indicating that shunt system works well and outcomes could be examined. In the case of non-responders, there were two possibilities. One is a malfunction of shunt system such as partial valve failure, etc. Another is a normal shunt system did not improve the symptoms. In most cases, we did not discriminate these two, indicating that non-responders are not a suitable group for interventional study. In describing post-operative study, we had better to use a term of “responder” instead of “iNPH.” Accordingly, we replace the term in result and discussion sections, and describe the responder as follows.

Line 853~856: By the surgical intervention, clinical symptoms are gradually attenuated in the shunt responders, in which shunt system functions well (23). In the present study, the responders were examined for their marker levels in CSF before and after the operation.  

Comment #7

The Introduction could be improved by including data relating to the biological significance of the CSF proteins measured in this paper.

Response to comment #7

I agree with the reviewer. For describing the biological significance of the CSF proteins, the following sentences were inserted in introduction section.

Line 83~88 for Tf function: Iron plays many important roles in the brain, including involvement in myelination, neurotransmission and electron transfer in the respiratory chain (8). However, free iron results in cellular toxicity by generating harmful reactive oxygen species (ROS) (9) . Iron, therefore, needs to bind to specific carrier proteins such as transferrin (Tf) in the extracellular fluid and blood (10).

Line 235-239 for PGDS function: Another major glycoprotein in CSF is lipocalin-type prostaglandin D2 synthase (L-PGDS). Urade et al. reported that a major CSF glycoprotein, L-PGDS is biosynthesized by leptomeninges and the choroid plexus, CSF-producing tissue (13). The molecule is a member of a su­perfamily of lipocalins that have diverse physiological functions, including binding and transporting lipophilic compounds, induction of sleep, etc (14).  

Comment #8

The discussion section could be improved by analyzing the biological implications of the study findings.

Response to comment #8

In addition to application of Man-Tf to SIH diagnosis, we discuss its possible function as follows.

Line 920-924: Tf captures ferric iron with very high affinity, Kd = 1 x 10-20, which prevents Fe-dependent cellular toxicity. Indeed, Fe toxicity is suspected to be responsible for neuron death in AD. Man-Tf, mainly produced by neurons, may play a role in local protective protein against iron toxicity, although its controlling mechanisms have not been elucidated yet.

Reviewer 4 Report

The authors of this manuscript have previously shown that levels of mannose-terminated glycans (Man-Tf) increase in the cerebral spinal fluid (CSF) of Alzheimer’s disease (AD) patients. Here, they want to see if this occurs in other neurological diseases, including depression, schizophrenia, spontaneous intracranial hypotension (SIH), and idiopathic normal pressure hydrocephalus (iNPH), and also if this increase of Man-Tf levels correlates to an increase in lipocalin-type prostaglandin D2 synthase (PGDS) and N-acetylglucosame-terminated glycans (GlcNAc-Tf). They find that Man-Tf, PGDS, and GlcNAc-Tf were significantly correlated in depression, schizophrenia, SIH, and iNPH. Though in the end, they only conclude that Man-Tf, in addition to AD, could be a good diagnostic marker for SIH.

The findings in this paper are novel, though it was very hard to tease out what their main conclusions were at the end and why their findings could be useful. I recommend the paper for publication with the following revisions.

1.)  They go through a lot of data to show that Man-Tf, PGDS, and GlcNAc-Tf were significantly correlated in depression, schizophrenia, SIH, and iNPH.  Though in the end they only conclude that Man-Tf is diagnostically relevant for SIH.  Is the correlation at all important if PGDS and GlcNAc turn out to be diagnostically irrelevant?

2.)  There is only a diagnostic section for SIH, how do they conclude that Man-Tf, PGDS, and GlcNAc-Tf are not diagnostically relevant for depression and schizophrenia?

3.)  The relevance of the post-operative data is not well explained. It is hard to understand why this is presented.

4.)  Last two paragraphs summarize/discuss their previous paper at too great a depth. This space would be better used better explaining the relevance of their current data. 

Author Response

Comments and Suggestions for Authors

The authors of this manuscript have previously shown that levels of mannose-terminated glycans (Man-Tf) increase in the cerebral spinal fluid (CSF) of Alzheimer’s disease (AD) patients. Here, they want to see if this occurs in other neurological diseases, including depression, schizophrenia, spontaneous intracranial hypotension (SIH), and idiopathic normal pressure hydrocephalus (iNPH), and also if this increase of Man-Tf levels correlates to an increase in lipocalin-type prostaglandin D2 synthase (PGDS) and N-acetylglucosame-terminated glycans (GlcNAc-Tf). They find that Man-Tf, PGDS, and GlcNAc-Tf were significantly correlated in depression, schizophrenia, SIH, and iNPH. Though in the end, they only conclude that Man-Tf, in addition to AD, could be a good diagnostic marker for SIH.

The findings in this paper are novel, though it was very hard to tease out what their main conclusions were at the end and why their findings could be useful. I recommend the paper for publication with the following revisions.

General response

Before addressing the reviewer’s comments one by one, I would like to mention critical issues as follows. In the present study, diagnostic accuracy mainly depends on difference of marker levels between patient and control groups. If the two groups show the same marker levels, the marker could not differentiate the patients at all. Even in this case, markers could be correlated each other. This indicates that correlation is an independent parameter from diagnostic accuracy. The aim of this study is to examine correlations of the brain-derived proteins, in CSF of neurological diseases. The present and previous study revealed that the markers are well correlated despite of their different origin in various neurological diseases. The findings provide basic knowledge about metabolism of CSF and its solute. I believe that the finding is worth to be reported even if the markers are not applicable to clinical medicine. Of course, it is important to examine whether the brain-derived proteins could be a diagnostic marker or not. Indeed, we searched for new markers based on criteria as follows: more than 85% sensitivity and 85% specificity together with more than 0.9 of area under the ROC curve (AUC). It is notable that each marker should be examined based on the above criteria because the maker correlation is independent parameter from diagnostic accuracy. For example, Man-Tf is a good marker for diagnosing SIH with 97% sensitivity, 95% specificity, and 0.98 AUC. In contrast, both of GlcNAc-Tf and PGDS show 77% sensitivity and 75% specificity, respectively, and their AUCs are 0.59 and 0.78, respectively. Thus, Man-Tf is a much better marker than GlcNAc-Tf and PGDS. In addition, we compared marker levels in other disease patients with those in controls (100%). Depression shows relative levels of Man-Tf, GlcNAc-Tf, and PGDS being 125%, 112%, 148%, respectively, while schizophrenia shows relative levels of Man-Tf, GlcNAc-Tf, and PGDS being 113%, 107%, 139%, respectively. The slight differences of marker levels were evident even in Figure 1, indicating that these markers are obviously poor markers for diagnosing the psychological diseases. I agree with the reviewer that search for a new marker is important especially when marker levels are significantly different between patients and controls. Levels of GlcNAc-Tf and Man-Tf in iNPH were significantly higher than those in non-iNPH. Therefore, markers were examined for their diagnostic accuracy and the results were inserted as a new Table 8 and line 375~380.

Comment (1)

They go through a lot of data to show that Man-Tf, PGDS, and GlcNAc-Tf were significantly correlated in depression, schizophrenia, SIH, and iNPH. Though in the end they only conclude that Man-Tf is diagnostically relevant for SIH. Is the correlation at all important if PGDS and GlcNAc turn out to be diagnostically irrelevant?

Response to Comment (1)

We believe that the marker correlation itself is a novel finding and worth to be reported, even if some markers are not relevant to diagnosis. In the present study, marker correlations are independent parameter from diagnostic accuracy. I agree with the reviewer that search for other markers are important especially when marker levels are significantly different between patients and controls. Levels of GlcNAc-Tf and Man-Tf in iNPH were significantly lower than those of non-iNPH. Therefore, markers were examined for their diagnostic accuracy. GlcNAc-Tf and Man-Tf showed 62% and 72% sensitivity, 38% and 28% specificity, and 0.55 and 0.65 of AUC, respectively. The results for iNPH diagnosis were inserted as a new Table 8 and line 375~380. 

Comment (2)

There is only a diagnostic section for SIH, how do they conclude that Man-Tf, PGDS, and GlcNAc-Tf are not diagnostically relevant for depression and schizophrenia?

Response to Comment (2)

In the present study, significant difference in marker levels are most important parameters for differentiating disease patients and controls. As shown in Figure 1, Man-Tf, PGDS, and GlcNAc-Tf does not show significant difference between controls and psychological diseases, indicating that they are not good markers. Only when marker levels were significantly different, we examined diagnostic accuracy of markers. We define criteria for good markers as follows: more than 85% sensitivity and 85% specificity together with more than 0.9 of area under the ROC curve (AUC). The sentence was inserted in line 363~365. I agree with the reviewer that only a diagnostic section for SIH may be insufficient. I added a new Table 8 and line 375~380 for iNPH diagnosis, but no marker had good diagnostic accuracy.

Comment (3)

The relevance of post-operative data is not well explained. It is hard to understand why this is presented.

Response to Comment (3)

As the reviewer pointed out, we should describe the relevance of post-operative experiments prior to present the data. We inserted the following sentences in line 392~396: After the operation, symptoms such as dementia and gate disturbance are gradually attenuated (shunt responder), suggesting that shunt system functions well and CSF metabolism are normalized. The responders were examined for their marker levels with the expectation that the operation changes the marker levels in the responders.

I have a comment on design of clinical study as follows.

There are two kind of clinical studies; observational and interventional studies.

In observational study, researchers observe and track health outcomes over time. No attempt is made to affect the outcome. In interventional study, participants are treated with medication or operation and certain outcomes are measured. In general, interventional study is regarded as showing higher evidence level. In the present study, marker analyses on CSF of depression, schizophrenia, SIH, and iNPH were observational study. In contrast, marker analysis on CSF of post-operative patients (shunt responders) is an interventional study, in which marker levels were increased and correlated even after shunt operation.

Comment (4)

Last two paragraphs summarize/discuss their previous paper at too great a depth. This space would be better used better explaining the relevance of their current data.

Response to Comment (4)

I agree with the reviewer that paragraphs for summarizing and discussing our previous paper is too much and inappropriate for the present discussion section. We removed this portion and biological implications of the present findings.
